# Improving Guideline-Recommended Colorectal Cancer Screening in a Federally Qualified Health Center (FQHC): Implementing a Patient Navigation and Practice Facilitation Intervention to Promote Health Equity

**DOI:** 10.3390/ijerph21020126

**Published:** 2024-01-24

**Authors:** Kathryn M. Glaser, Christina R. Crabtree-Ide, Alyssa D. McNulty, Kristopher M. Attwood, Tessa F. Flores, Allana M. Krolikowski, Kevin T. Robillard, Mary E. Reid

**Affiliations:** 1Department of Cancer Prevention and Populations Science, Roswell Park Comprehensive Cancer Center, Buffalo, NY 14263, USA; alyssa.mcnulty@roswellpark.org; 2Department of Medicine, Roswell Park Comprehensive Cancer Center, Buffalo, NY 14263, USA; christina.crabtree-ide@roswellpark.org (C.R.C.-I.); tessafaye.flores@roswellpark.org (T.F.F.); mary.reid@roswellpark.org (M.E.R.); 3Department of Biostatistics & Bioinformatics, Roswell Park Comprehensive Cancer Center, Buffalo, NY 14263, USA; kristopher.attwood@roswellpark.org; 4Jericho Road Community Health Center, Buffalo, NY 14213, USA; allana.krolikowski@jrchc.org; 5Department of Internal Medicine, Roswell Park Comprehensive Cancer Center, Buffalo, NY 14263, USA; kevin.robillard@roswellpark.org

**Keywords:** patient navigation, practice facilitation, colorectal cancer screening, outreach and education, refugee and immigrant health, limited English proficiency (LEP)

## Abstract

Background: Colorectal cancer (CRC) screening is effective in the prevention and early detection of cancer. Implementing evidence-based screening guidelines remains a challenge, especially in Federally Qualified Health Centers (FQHCs), where current rates (43%) are lower than national goals (80%), and even lower in populations with limited English proficiency (LEP) who experience increased barriers to care related to systemic inequities. Methods: This quality improvement (QI) initiative began in 2016, focused on utilizing patient navigation and practice facilitation to addressing systemic inequities and barriers to care to increase CRC screening rates at an urban FQHC, with two clinical locations (the intervention and control sites) serving a diverse population through culturally tailored education and navigation. Results: Between August 2016 and December 2018, CRC screening rates increased significantly from 31% to 59% at the intervention site (*p* < 0.001), with the most notable change in patients with LEP. Since 2018 through December 2022, navigation and practice facilitation expanded to all clinics, and the overall CRC screening rates continued to increase from 43% to 50%, demonstrating the effectiveness of patient navigation to address systemic inequities. Conclusions: This multilevel intervention addressed structural inequities and barriers to care by implementing evidence-based guidelines into practice, and combining patient navigation and practice facilitation to successfully increase the CRC screening rates at this FQHC.

## 1. Introduction

Colorectal cancer (CRC) screening is effective in both the prevention and early detection of colon cancer, yet CRC continues to be one of the most common cancers diagnosed in both men and women in the United States (US), and a leading cause of cancer-related deaths [1]. Populations served by Federally Qualified Health Centers (FQHCs) in the US are screened at lower rates than the general population for all major cancers, including colorectal cancer, and screening rates are even lower in populations with limited English proficiency (LEP) [2,3]. The national CRC screening average for FQHCs is 46%, while the national goal for the general population is 80%, indicating a significant disparity in screening rates in populations served by FQHCs for patients who experience increased barriers to care and health system inequities, with the lowest screening rates in patients with LEP (34%) [4].

Although FQHCs are motivated to promote CRC screenings and demonstrate improvement as a clinical quality measure (CQM), discussions around cancer screening and prevention are challenging because of the time constraints on care teams handling the complex needs of patients served, compounded by the language and economic barriers facing underserved populations [4,5]. It is well established that provider recommendation is one of the greatest motivators for cancer screening, especially in English-proficient (EP) populations, although, despite the intent to screen patients, several barriers limit these discussions between providers and patients [6]. Time constraints during office visits, insufficient access to colonoscopy resources and facilities based on capacity and health insurance, patient follow-through on recommendations, and patients’ fear and misunderstanding regarding colonoscopy preparation and procedure are just some of the challenges FQHCs face when promoting CRC screening [7]. These issues, especially those related to systemic inequities in the US healthcare system, are compounded by LEP and cultural barriers [6,7].

In 2018, the American Cancer Society (ACS) pioneered lowering the recommended screening age from 50 to 45 years of age, which intended to shift diagnosis to a younger age and address disparities in CRC incidence across racial subgroups [1]. Colonoscopy is the only screening option that allows for the removal of precancerous lesions at the same time and is credited for the shift in detection to earlier-stage disease and a steep decline in CRC incidence [1]. Since it has a greater capacity for CRC prevention than other tests, the dissemination of colonoscopy screening is preferred and is considered the gold standard. Additionally, the screening interval is 10 years for average-risk individuals compared to annual testing using stool-based tests, such as fecal immunochemical testing (FIT).

Despite these improvements, not all populations have benefited equally, and screening targets remain unmet in underserved populations across the US, including under-resourced areas of western New York (WNY) [4,8,9,10,11,12,13]. In the US, National Health Center Program awardees and look-alikes recognized through Health Resources and Services Administration (HRSA) are funded by the federal government and are required to report a core set of clinical quality measures based on current US Preventive Services Task Force guidelines. These clinical quality measures include CRC screening rates as part of a standardized reporting system, known as the UDS, and are publicly reported on the HRSA website [4]. Average CRC screening rates at FQHCs nationally were 44% in 2018 and 42.8% in 2022, compared to the overall national average of 59%, which are well below the National Colorectal Cancer Roundtable’s target of 80% CRC screening by 2018 and the HealthyPeople 2030 target of 74.4% of adults receiving guideline-recommended screening [1,14,15,16,17]. This lack of CRC screening contributes to the cancer disparities in underserved, low-income patients, especially in immigrant populations with LEP [18,19,20,21].

Barriers to screening continue to persist, leading to health inequity. Low education, low health literacy, low income, and medical mistrust are consistently reported barriers to adequate CRC screening, especially among underserved populations and racial/ethnic minority populations [2,12,22,23]. Additionally, racial and ethnic disparities in screening and, importantly, cancer mortality, are well-documented [12,21,22,24,25,26]. Insurance coverage and poor access to primary care providers are known barriers to cancer screening [19,22,23,25,27,28]. These barriers are compounded by social and structural drivers of health for patient populations in urban FQHCs in WNY, where patient populations are largely low-income, have public insurance, and have low health literacy. Addressing these barriers to achieve health equity has historically required specific intervention strategies and are challenging to overcome [4].

In Buffalo, New York (NY), the foreign-born population has been rising steadily for over 15 years, with now more than 10.4 percent of the city composed of recent immigrants and refugees with LEP, many from Burma, Nepal, Sudan, Somalia, and Vietnam, in addition to an established Puerto Rican population [29]. In 2016, Roswell Park Comprehensive Cancer Center (Roswell), an NCI-designated cancer center in downtown Buffalo, partnered with the Jericho Road Community Health Center (JRCHC) to increase CRC screening rates as a quality improvement (QI) intervention. The JRCHC is an urban FQHC in Buffalo with a culturally and linguistically diverse patient population (over 50% with LEP) and which represent over 60 different language groups living in one of the poorest geographic regions in the city [29]. The JRCHC is a Patient-Centered Medical Home (PCMH) and holds the highest recognition from the National Committee for Quality Assurance (NCQA). They offer full-spectrum care to all, regardless of insurance status or ability to pay. The JRCHC continues to grow and additionally supports multiple clinics worldwide in Sierra Leone, Nepal, and the Democratic Republic of Congo, seeking to improve healthcare for the most vulnerable individuals in the home countries of many resettled refugees in Buffalo.

Prior to this partnership, JRCHC patients referred for colonoscopy received appointments and instructions in English from screening facilities. Given the low literacy and low health literacy levels, compounded by language barriers, this process contributed to frequently missed appointments, as well as patients arriving unaware and unprepared for the colonoscopy procedure itself. In response, screening facilities in the community limited the number of colonoscopy procedures available to JRCHC patients, furthering the inequity in access. Additionally, many facilities developed a no-show or late-cancellation policies that included fees to the patient that are not covered by insurance. In efforts to resolve these barriers, the JRCHC approached Roswell for potential solutions. Using a community-based participatory approach, we collectively designed and developed an intervention to improve the CRC screening rates.

Our primary aim was to address these barriers to care and inequity in access to CRC screening, using a combined model of practice facilitation and patient navigation as a targeted intervention. Our assumption was that this intervention would successfully increase the CRC screening rates at this FQHC, acting as an important step to addressing health disparities and health equity, specifically in patients with LEP. Practice facilitation is broadly defined as an effective strategy to improve primary healthcare processes and outcomes. Practice facilitation uses quality improvement (QI) and practice improvement approaches and methods to build the internal capacity of primary care practices to help engage in improvement activities over time and support reaching improvement goals [30,31,32,33,34]. Practice facilitation provides resources to support the dissemination and implementation of evidence-based guidelines, in addition to QI approaches to performance improvement, including audit and feedback, the identification and spread of best practices, and academic detailing to improve the quality in primary care settings [35]. Practice facilitation and patient navigation have been identified as intervention strategies that have the potential to increase cancer screening rates and have demonstrated success in other populations and settings [35,36,37].

Between August 2016 and December 2018, we pilot tested a QI intervention in one of two JRCHC clinic sites, one serving predominately patients with LEP (intervention) and the other majority Black or African American (control). Since the time of the intervention, the JRCHC has now grown to five clinic locations in Buffalo, including one within a homeless shelter, supporting the broader community in the Buffalo area. Based on the success of the initial pilot and clinical expansion at the JRCHC, this collaboration has expanded since December 2018, now supporting two full-time patient navigators working with all clinic sites and eligible patients.

## 2. Methods

The initial QI intervention started in August 2016 with the goal of increasing the CRC screening rates at one of JRCHC’s clinic sites with the highest LEP population. We measured the CRC screening rates at four time points from August 2016 to December 2018, comparing the two clinic locations (the intervention and control sites). At the intervention site (Site 1), we used patient navigation combined with practice facilitation as an intervention strategy to increase the adherence to CRC screening guidelines (colonoscopy or stool-based testing). At Site 2 (the control site), there was no intervention. CRC screening registries were developed based on CQM requirements (active patients, men and women, ages 50–75).

In an effort to address barriers to care and health system inequities, a full-time patient navigator was hired by Roswell and embedded in the JRCHC to provide both provider and patient education, develop culturally tailored patient education materials (visual), schedule/coordinate services (transportation, interpreters, obtaining prep solution for colonoscopy), and distribute stool-based tests using FIT for those deferring or ineligible for colonoscopy as a screening option. Patients were identified by several methods, including CRC screening registries for all eligible patients, direct provider referral, daily appointment schedules, open CRC screening order reports, and incentives based on insurance plans. Once patients were contacted, the patient navigator provided culturally tailored education modified to the overall literacy and health literacy levels. Patients who agreed to screening after the education session were scheduled for a colonoscopy or given a FIT test, and barriers to care (e.g., language and transportation) were addressed to ensure completion of the CRC screening.

As part of this tailored intervention, the patient navigator provided reminder calls prior to colonoscopy procedures at 1 week, 2 days (for prep), and 1 day before to avoid no-shows and last-minute cancellations. If rescheduling was required, endoscopy facilities were notified at a minimum of 24 h in advance to avoid additional patient fees, improve relationships with community partners, and increase efficiency within the healthcare system. Patients electing to complete the FIT tests received the test kit, detailed instructions, and follow-up phone calls for up to 3 weeks until completion was verified by obtaining test results and patient navigators communicated the results with PCPs.

Patients without health insurance were referred to the New York State (NYS) Cancer Services Program to facilitate CRC screening. This unique program provides breast, cervical, and CRC screening for uninsured NYS residents at no cost. CRC screening is performed utilizing a FIT test, and then colonoscopy for a subsequent positive FIT. Additionally, if cancer is detected, uninsured patients are covered through an NYS health insurance program to cover costs related to cancer care. This partnership allows our program to provide CRC screening to any age-eligible member in our community.

Throughout the implementation of this intervention, barriers and facilitators were identified, and we adapted the intervention accordingly to meet the needs of this diverse patient population using a continuous QI methodology. This included the development of visual aids for patient education in the early stages of the intervention, such as the use of animal and human anatomy to review the gastrointestinal (GI) organs using pictures (Figure 1) and videos (https://www.youtube.com/watch?v=2ML78wLmVaA, accessed on 3 January 2024)). This strategy was in response to low health literacy as well as cultural and linguistic variation in explaining and understanding the GI system [37]. These visual aids helped educate on the importance of colon health and CRC screening, as well as what to expect and how to prepare for a colonoscopy.

To optimize the navigation process, patient navigators were trained in all relevant applications. This included EMR systems, insurance verification and prior authorization, as well as health information exchanges, registration, scheduling, and the coordination of supportive services (interpretation and transportation). The goal was to optimize the efficiency, maintain good working relationships between community partners, and reduce no-shows and subsequent waste in the healthcare system.

For the pilot intervention, overall patient characteristics were summarized by site using (1) mean, median, standard deviation, and range for continuous measures, and (2) frequencies and relative frequencies for categorical variables. Comparisons were made using the Mann–Whitney U and Fisher’s exact tests, as appropriate. Odds ratios for comparing screening rates between interventions and control sites at each time-point were estimated, along with corresponding 95% confidence intervals. Screening rates were modeled as a function of the site and time, and their two-way interaction using a logistic regression model. Wald-type chi-square tests were used to evaluate the main and interaction effects. All analyses were conducted in SAS v9.4 (Cary, NC, US) at a nominal significance level of 0.05. No imputation methods were used for missing data and no adjustments were made for multiple testing. The Institutional Review Board at Roswell determined that this QI project did not meet the definition of human subjects research.

## 3. Results

Between August 2016 and December 2018, CRC screening rates at the intervention site significantly increased over time from 31% to 59% (*p* < 0.001), with the most notable change in patients with LEP, providing evidence to support the effectiveness of this combined intervention of practice facilitation and patient navigation. Patient characteristics from both clinical sites and the overall study population are summarized in Table 1. There were 2399 patients eligible for CRC screening at the intervention site and 935 total patients eligible at the control site. Throughout the course of the intervention, 780 patients were screened from the intervention site compared to 280 at the control site. Based on patient preference and/or insurance, patients were screened by endoscopists in the community and at Roswell.

The intervention site had a greater proportion of patients with LEP (58%), with only 42% speaking English as a first language, compared to 73% primarily English-speaking at the control site. Both clinic sites had patients with predominantly government-issued (public) insurance coverage (63%); however, more patients at the control site were covered by commercial insurance than the intervention site (27% versus 19%, respectively; *p* < 0.001). There were no significant differences in age, gender, or prior CRC screening rates between the intervention and control sites.

Changes in the colonoscopy screening rates by site over time are shown in Figure 2. At baseline, the screening rates at the intervention site (Site 1) were essentially the same as the control site (Site 2). Significantly more procedures occurred at the intervention site compared to the control, with a significant time–site interaction (*p* < 0.001), where screening rates demonstrated greater improvement in the intervention site. While there was no difference in the screening rates at baseline, there was a marked increase in the screening rates by 2017 (*p* < 0.001), which extended through the 2018a (*p* < 0.001) and 2018b (*p* < 0.001) time-points.

The stratum-specific odds ratios comparing screening at the intervention site compared to the control site by year are presented in Table 2, separated by colonoscopy and FIT. At baseline (2016), there were no differences in colonoscopy screening between the two sites. By the end of 2018, patients at the intervention site had two-fold higher odds of colonoscopy screening compared to the control site (OR = 1.94; 95% CI = 1.61, 2.35). For the FIT testing, the intervention site at baseline showed 60% higher odds of screening at baseline. This difference was magnified; by the final year, there was a four-fold increase (OR = 4.22; 95% CI = 2.67, 6.65). We also assessed differences in the screening rates by demographic characteristics. We omitted odds ratios for groups with small sample sizes, specifically certain language groups, due to unstable estimates. Men and women both showed significant improvement in screening over time. Both men and women were twice as likely to have a colonoscopy in the intervention group, and four-times more likely to have a FIT test by the end of 2018, compared to men and women at the control site. Older subjects (=> 58 years) were more likely to have a colonoscopy than <58 years at both sites, and significantly more likely to be screened when seen at the intervention site (*p* < 0.0001).

Based on the success of this pilot intervention, efforts expanded by the end of 2018 to two full-time navigators supporting all clinics at the JRCHC. Overall, the CRC screening rates between 2018 and 2022 for the JRCHC are shown in Figure 3 based on the CQM reports for CRC screening through Health Center Program Uniform Data Systems (UDSs) reports. The CRC screening rates are compared to other FQHCs across NYS and the US based on averages (Figure 3). Since 2018, these UDS reports serve as the primary means for tracking increases over time and enhance the continuous QI process. As seen in Figure 3, the CRC rates have steadily increased from 43% to 50%, exceeding both NYS and US averages. Additionally, this demonstrates the continued and sustained effectiveness of patient navigation to address systemic inequities in CRC screening. As of January 2023, the CQM reports for CRC screening have lowered to 45 years old, adding another group of patients to be screened.

## 4. Discussion

Despite major barriers and inequities within the US healthcare system, this tailored intervention is a successful strategy to increase the CRC screening rates at this culturally and linguistically diverse FQHC, particularly in patients with LEP. Both provider and patient education, combined with practice facilitation and patient navigation, is critical to the adherence to national CRC screening guidelines. Many patients receiving care at FQHCs experience the extreme burden of social determinants of health, and may prioritize housing, food, and income security over screening. Adding patient navigation to the clinics significantly improved the successful completion of CRC screening by overcoming barriers to care and increasing health equity. Additionally, core members of the team meet quarterly to review the QI reports and continue to enhance the program, identifying any subpopulations needing a different strategy as part of the continuous QI process.

Tailoring the intervention to the patient population with LEP was a successful strategy to improving screening rates and building a sustainable program. Patient navigators developed visual aids for addressing LEP and low health literacy, which improved the understanding of preparing for a colonoscopy [38]. Having dedicated staff and culturally tailored education materials increased the screening rates while increasing the efficiency of the screening procedures to maximize healthcare utilization and reduce waste. Embedding staff within the primary care office allowed for critical hand-offs to occur and follow through with provider recommendations for CRC screening tests. Study limitations include the differences in the two clinical sites based on the language and racial breakdown. Although there was a significant increase in the LEP population and when the intervention was later applied to all clinic sites, the screening rates continued to increase across both sites overall. This model not only improved the screening rates, but also quality of care, and navigators continued to have a patient-centered approach.

This study suggests this is a successful intervention strategy for addressing health equity and improving the CRC screening rates in FQHCs through assistance with patient identification, tailored provider and patient education, and the mitigation of barriers to screening (e.g., transportation, interpreters, time off work, and obtaining scripts), especially among immigrants and refugees with LEP. The intervention supported patients and providers in overcoming known barriers by embedding dedicated patient navigators in the clinic [39]. Implementing this dedicated staffing model improved adherence, decreased no-show rates, and reduced the number of patients lost to follow-up care.

This project, grounded in both health services research and clinical care by implementing evidence-based guidelines into practice, worked directly with underserved communities in our area to address health inequities and provides a model for other QI initiatives. This further supports the evidence demonstrated by prior studies that patient navigation and facilitation are successful intervention strategies for improving CRC screening rates, particularly in safety-net practices [37,39]. The success of the pilot intervention was key to the long-term sustainability and resulted in additional support for our navigation program from our cancer center through strategic business plans to expand to other FQHCs regionally, and also securing grant funds for similar cancer screening initiatives.

## 5. Conclusions

Although this intervention focused on CRC screening, the opportunity for expansion into other screening services is clear. Patients also benefit from navigation for other cancer screening, such as breast and lung cancer screening. Tailored expansion and testing into other local settings for cancer screening navigation is an ideal next step for this type of program to improve the CRC screening rates across the community rather than within a single FQHC population. Additionally, patient navigation interventions have the potential to reduce lost revenue from missed/underprepared procedures. It also has the potential to reduce the overall healthcare costs related to the burden of cancer treatment by navigating patients to colonoscopy for early detection and cancer prevention by removing precancerous polyps, preventing cancer from occurring. Further investigation into cost savings may promote this model of intervention in other, more aggressive forms of cancer, such as lung cancer screening.

This multilevel approach to addressing barriers to care and health equity through an effective intervention that implemented evidence-based guidelines into practice, and combined patient navigation and practice facilitation, successfully increased the CRC screening rates at this FQHC to well above the state and national averages. In addition, this model of patient navigation can be implemented in other regions to address important health disparities and increase health equity. Partnerships between academic medical centers and FQHCs provides a quality improvement infrastructure for multiple initiatives, especially those related to required quality measures. Enhancing these partnerships provides a successful model for increasing cancer screening rates and could be generalized for other preventative services or the management of chronic conditions. Further research into the elements of navigation and practice facilitation might lead to even more improvements in screening compliance in challenging populations.

## Figures and Tables

**Figure 1 ijerph-21-00126-f001:**
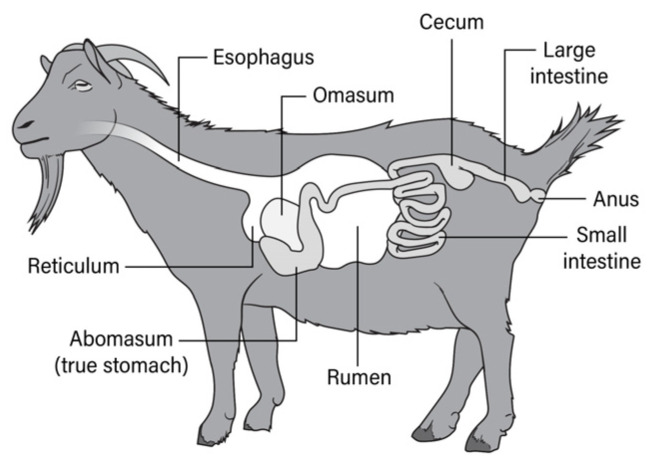
Goat digestive system.

**Figure 2 ijerph-21-00126-f002:**
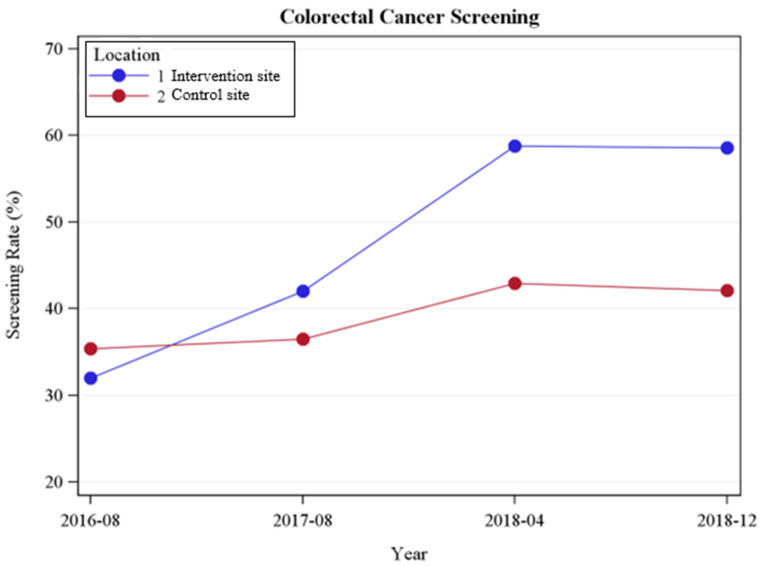
Changes in colonoscopy screening by site at the FQHC (August 2016 to December 2018).

**Figure 3 ijerph-21-00126-f003:**
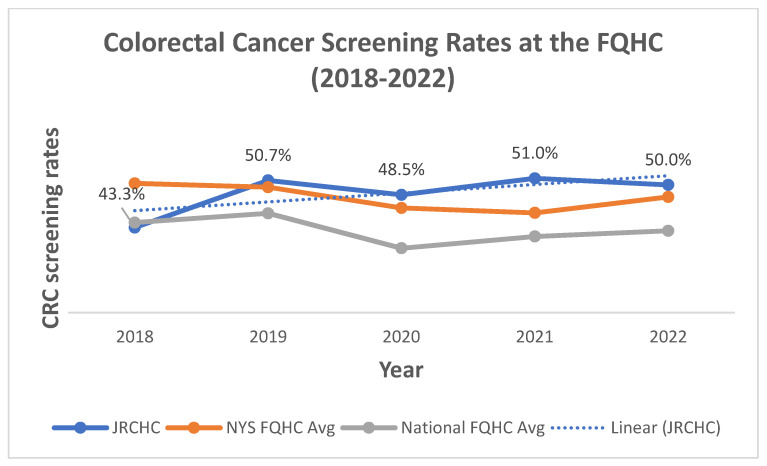
Changes in colonoscopy screening at the FQHC (all sites) from 2018 to 2022.

**Table 1 ijerph-21-00126-t001:** Baseline demographic characteristics at the FQHC for all clinical sites across all years of the intervention (August 2016 to December 2018).

Variable	Level	Site 1 (Intervention)	Site 2 (Control)	Overall	*p*-Value
N		2399	935		
Age	Mean (SD)	58.4 (6.8)	57.7 (6.2)	58.2 (6.7)/3334	0.05
	Median (Range)	57 (50, 75)	57 (50, 75)	57 (50, 75)	
Age Group	=>58	1229 (51.2%)	510 (54.5%)	1739 (52.2%)	0.09
	<58	1170 (48.8%)	425 (45.5%)	1595 (47.8%)	
Sex	Female	1252 (52.2%)	464 (49.6%)	1716 (51.5%)	0.18
	Male	1147 (47.8%)	471 (50.4%)	1618 (48.5%)	
Language	Arabic	152 (6.6%)	15 (1.7%)	167 (5.2%)	<0.001
	Bangla/Bengali	17 (0.7%)	150 (16.5%)	167 (5.2%)	<0.001
	Burmese	371 (16.1%)	10 (1.1%)	381 (11.9%)	<0.001
	English	958 (41.7%)	660 (72.6%)	1618 (50.4%)	<0.001
	Nepali	217 (9.4%)	2 (0.2%)	219 (6.8%)	<0.001
	Other	177 (7.7%)	31 (3.4%)	208 (6.5%)	<0.001
	Somali	110 (4.8%)	7 (0.8%)	117 (3.6%)	<0.001
	Spanish	224 (9.7%)	13 (1.4%)	237 (7.4%)	<0.001
	Swahili	42 (1.8%)	1 (0.1%)	43 (1.3%)	<0.001
	Vietnamese	32 (1.4%)	20 (2.2%)	52 (1.6%)	0.12
	Non-English	1342 (58.3%)	249 (27.4%)	1591 (49.6%)	-
Insurance	Combined	342 (14.9%)	109 (12.1%)	451 (14.1%)	0.04
	Commercial	423 (18.5%)	240 (26.6%)	663 (20.8%)	<0.001
	Government	1463 (63.9%)	532 (59.0%)	1995 (62.5%)	0.01
	Other	61 (2.7%)	20 (2.2%)	81 (2.5%)	0.53
Prior CRC Screening	No Yes	1657 (69.1%) 742 (30.9%)	670 (71.7%) 265 (28.3%)	2327 (69.8%) 1007 (30.2%)	0.14

**Table 2 ijerph-21-00126-t002:** Stratum-specific odds ratios (95% confidence interval) for comparison of screening rates at the intervention site versus the control site at the FQHC (August 2016 to December 2018).

	OR (95% CI)			
	2016	2017	2018a	2018b
**Colonoscopy Screening**				
Overall	0.86 (0.69, 1.06)	**1.26 (1.04, 1.53)**	**1.89 (1.52, 2.35)**	**1.94 (1.61, 2.35)**
Insurance Source:				
Combined	0.64 (0.35, 1.16)	1.10 (0.64, 1.88)	1.43 (0.79, 2.61)	**2.36 (1.33, 4.21)**
Commercial	0.76 (0.49, 1.18)	**1.84 (1.16, 2.92)**	1.64 (0.95, 2.82)	**1.93 (1.25, 3.00)**
Government	0.96 (0.72, 1.27)	1.22 (0.96, 1.55)	**2.08 (1.60, 2.72)**	**2.04 (1.61, 2.59)**
Other Insurance	1.69 (0.31, 9.21)	0.17 (0.03, 1.07)	1.08 (0.21, 5.45)	0.64 (0.17, 2.39)
Female	0.94 (0.70, 1.27)	1.30 (0.99, 1.69)	**1.84 (1.37, 2.47)**	**1.81 (1.39, 2.37)**
Male	0.77 (0.57, 1.05)	1.22 (0.92, 1.62)	**1.95 (1.41, 2.68)**	**2.09 (1.59, 2.74)**
Language:				
Burmese	1.12 (0.15, 8.20)	1.23 (0.24, 6.31)	3.18 (0.67, 14.96)	1.53 (0.25, 9.52)
English	0.66 (0.50, 0.86)	0.82 (0.64, 1.05)	**1.34 (1.01, 1.76)**	**1.50 (1.19, 1.91)**
Nepali	-#	-	-	-
Other	1.13 (0.70, 1.84)	**2.07 (1.32, 3.25)**	**2.47 (1.58, 3.86)**	**2.77 (1.82, 4.21)**
Spanish	2.10 (0.43, 10.29)	0.89 (0.25, 3.21)	1.32 (0.18, 9.68)	1.04 (0.17, 6.39)
Non-English	1.47 (0.95, 2.26)	**2.68 (1.81, 3.96)**	**3.13 (2.09, 4.67)**	**3.40 (2.32, 4.97)**
**FIT Testing Rates**				
Overall	**1.60 (1.08, 2.37)**	**1.80 (1.25, 2.59)**	**4.00 (2.49, 6.42)**	**4.22 (2.67, 6.65)**
Insurance				
Combined	1.97 (0.65, 5.90)	3.14 (0.93, 10.66)	2.09 (0.77, 5.68)	**4.66 (1.09, 19.96)**
Commercial	1.71 (0.67, 4.34)	1.78 (0.72, 4.40)	0.85 (0.31, 2.33)	**4.50 (1.34, 15.14)**
Government	1.46 (0.89, 2.39)	**1.78 (1.14, 2.78)**	**8.38 (3.88, 18.12)**	**4.80 (2.70, 8.55)**
Other Insurance	1.87 (0.19, 18.25)	-	0.26 (0.01, 4.68)	0.36 (0.06, 2.07)
Female	1.59 (0.92, 2.74)	**2.59 (1.51, 4.46)**	**3.22 (1.81, 5.73)**	**4.13 (2.13, 7.97)**
Male	1.60 (0.91, 2.82)	1.22 (0.74, 2.02)	**5.74 (2.46, 13.37)**	**4.34 (2.31, 8.17)**
Language:				
Burmese	-	0.91 (0.10, 8.21)	1.84 (0.21, 15.97)	-
English	0.95 (0.55, 1.65)	0.75 (0.45, 1.26)	**3.04 (1.45, 6.39)**	**2.03 (1.16, 3.55)**
Nepali	-	-	-	-
Other	1.65 (0.82, 3.32)	**2.42 (1.13, 5.18)**	**2.54 (1.26, 5.11)**	**7.02 (2.53, 19.42)**
Spanish	-	-	-	-
Non-English	1.66 (0.84, 3.28)	**2.78 (1.39, 5.59)**	**2.55 (1.34, 4.86)**	**7.31 (2.67, 20.02)**

# Not sufficient numbers in Site 2 for comparison; reference site = Site 2 (control site); FIT = fecal immunochemical testing; bolded numbers indicate statistical significance.

## Data Availability

The data presented in this study are available on request from the corresponding author upon reasonable request.

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
