# Peer review of "Improving Guideline-Recommended Colorectal Cancer Screening in a Federally Qualified Health Center (FQHC): Implementing a Patient Navigation and Practice Facilitation Intervention to Promote Health Equity"

_ijerph, 2024, doi:10.3390/ijerph21020126_

Round 1

Reviewer 1 Report

Comments and Suggestions for Authors

The authors site the screening rate at FQHCs and compare to the national screening goal but they should compare like with like. What are actual screening rates in the wider population?  Based on figure 2, at 2018 rates at FQHCs look to be the same as the national average. Is this the overall national average or the national average among FQHCs?

In citing the target screening rate, it should be mentioned what the guidelines state about frequency of screening. For example, ACS suggests FOI every two years for average risk adults and colonoscopies every 10 years.

Page 3 lists a wide variety of activities that the patient navigators engage in. Is the approach standardized in any way? To what extent might it vary by the particular patient navigator, the particular circumstances of the clinic and its patients, and their interaction? The key question here is how the intervention approach would be expanded to other sites, and how much variation might one expect to observe in outcomes across different sites. In other words, is it possible to identify the key determinants of success?

The authors need to provide more clarity about the sample. While N is given, how are screening rates calculated? They are presented at four points in time but do they show the cumulative number of patients screened? Since guidelines do not recommend annual screening, is a person x counted in the calculation of screening rates each year and if so does the screening rate reflect whether the person was screened in the last two years? I’m not actually sure what is going on since screening rates in the final period actually look like they decline a bit for the control group, suggesting that these rates are not in fact cumulative.

Since colonoscopy is not regularly recommended for average risk adults, are calculated colonoscopy rates for those individuals deemed to be higher risk? Screening rates for colonoscopy should be restricted to the subset of people who are higher risk, if such information is available.

Author Response

The authors site the screening rate at FQHCs and compare to the national screening goal but they should compare like with like. What are actual screening rates in the wider population?  Based on figure 2, at 2018 rates at FQHCs look to be the same as the national average. Is this the overall national average or the national average among FQHCs?

Screening rates at FQHCs are compared to other FQHCs across the state and to national averages in order to compare like with like. The figure (now Figure 3) has been revised to clarify and reflect this in the figure and in the explanation on page 8. National screening rates for the overall population were added on pg 2.

In citing the target screening rate, it should be mentioned what the guidelines state about frequency of screening. For example, ACS suggests FOI every two years for average risk adults and colonoscopies every 10 years.

Guidelines on frequency was added on pg 2.

Page 3 lists a wide variety of activities that the patient navigators engage in. Is the approach standardized in any way? To what extent might it vary by the particular patient navigator, the particular circumstances of the clinic and its patients, and their interaction? The key question here is how the intervention approach would be expanded to other sites, and how much variation might one expect to observe in outcomes across different sites. In other words, is it possible to identify the key determinants of success?

Extensive details regarding the navigation process were added on pg 4 and address these questions.

The authors need to provide more clarity about the sample. While N is given, how are screening rates calculated? They are presented at four points in time but do they show the cumulative number of patients screened? Since guidelines do not recommend annual screening, is a person x counted in the calculation of screening rates each year and if so does the screening rate reflect whether the person was screened in the last two years? I’m not actually sure what is going on since screening rates in the final period actually look like they decline a bit for the control group, suggesting that these rates are not in fact cumulative.

Language was added on page 2 to discuss guidelines for CRC cancer and how FQHCs reports through the UDS system based on current US Preventive Services Task Force guidelines.

Since colonoscopy is not regularly recommended for average risk adults, are calculated colonoscopy rates for those individuals deemed to be higher risk? Screening rates for colonoscopy should be restricted to the subset of people who are higher risk, if such information is available.

Additional information regarding rationale for using colonoscopy was added to pg 2.

Reviewer 2 Report

Comments and Suggestions for Authors

The manuscript titled "Improving Guideline-Recommended Colorectal Cancer Screening in a Federally Qualified Health Center (FQHC): Implementing a Patient Navigation and Practice Facilitation Intervention to Promote Health Equity" presents a study on enhancing colorectal cancer (CRC) screening rates in a Federally Qualified Health Center (FQHC) through patient navigation and practice facilitation. This quality improvement initiative, initiated in 2016, aimed to address systemic inequities and barriers to care, particularly among patients with limited English proficiency (LEP). The study reported significant increases in CRC screening rates at the intervention site, demonstrating the effectiveness of the implemented strategies.

1.       It would be helpful to include a discussion on the long-term sustainability of these interventions and their applicability in other FQHCs with similar challenges.

2.       In Table 2, please calculate and list the OR value in different age groups, as well as Arabic, Bangli, Somali, Swahili, and Vietnamese.

3.       In Figure 2, please label the Y-axis. Please also consider using other software, to improve the quality of the figure.

4.       Consider adding a few more visual aids or diagrams to illustrate the intervention process and results for enhanced readability.

5.       Expanding on how these findings contribute to the broader field of health equity would further strengthen the conclusion.

Author Response

  1. It would be helpful to include a discussion on the long-term sustainability of these interventions and their applicability in other FQHCs with similar challenges.

Additional language to address sustainability was added on pg 10.

  1. In Table 2, please calculate and list the OR value in different age groups, as well as Arabic, Bangli, Somali, Swahili, and Vietnamese.

Additional language was added on pg 7.

  1. In Figure 2, please label the Y-axis. Please also consider using other software, to improve the quality of the figure.

Figure 2 is now Figure 3 and has been adjusted accordingly.

  1. Consider adding a few more visual aids or diagrams to illustrate the intervention process and results for enhanced readability.

Extensive details regarding the navigation process and additional illustrations were added on pg 4 and address these questions.

  1. Expanding on how these findings contribute to the broader field of health equity would further strengthen the conclusion.

Additional language to address sustainability was added on pg 10.

Reviewer 3 Report

Comments and Suggestions for Authors

ABSTRACT

1.     Clear.

KEYWORDS

1.     5 words will be enough I don’t think the authors need the words early detection of 33 cancer and limited english proficiency (LEP)

INTRODUCTION

1.     Reference 1 is from 2017 there is a lot of updated literature on colorectal cancer statistics.

2.     The introduction contains details that are not needed for the study, must be shortened, and hypotheses and aims are missed.

MATERIALS AND METHODS

1.     IRB information must be added if there is any.

RESULTS

1.     Clear.

DISCUSSION

1.     Authors must compare their results to other studies from different countries and populations as well as different screening programs.

2.     It has been known that cancer is due to the combination of genetics lifestyle and environmental factors that can play a crucial role in the cancer risk; the population in this study is heterogeneous and authors must subsequent their analysis according to rase.

3.     Can be improved. 

CONCLUSION

Better to be shortened.

Comments on the Quality of English Language

Good!

Author Response

ABSTRACT

  1. Clear.

KEYWORDS

  1. 5 words will be enough I don’t think the authors need the words early detection of 33 cancer and limited english proficiency (LEP)

Early detection and LEP were removed from key words.

INTRODUCTION

  1. Reference 1 is from 2017 there is a lot of updated literature on colorectal cancer statistics.

Reference 1 was updated to 2023.

  1. The introduction contains details that are not needed for the study, must be shortened, and hypotheses and aims are missed.

Aims and goals were added on pg 3.

MATERIALS AND METHODS

  1. IRB information must be added if there is any.

IRB information added on pg 5

RESULTS

  1. Clear.

DISCUSSION

  1. Authors must compare their results to other studies from different countries and populations as well as different screening programs.

Additional language added to discussion on 9. Since the focus was on US and foreign-born living in US, comparing to health care systems in other countries would present a challenge, but is an excellent idea for a future manuscript! Thank you!

  1. It has been known that cancer is due to the combination of genetics lifestyle and environmental factors that can play a crucial role in the cancer risk; the population in this study is heterogeneous and authors must subsequent their analysis according to rase.

Additional analysis would be hard to add for this submission, particularly given the quick turn around for edits. Again, would be an excellent idea for a future manuscript!

  1. Can be improved. 

Additional language on pg 9.

CONCLUSION

Better to be shortened.

Word count is now met but was initially under journal requirements. Further shortening the conclusion presents a challenge.